# Recent Insights into the Interplay of Alpha-Synuclein and Sphingolipid Signaling in Parkinson’s Disease

**DOI:** 10.3390/ijms22126277

**Published:** 2021-06-11

**Authors:** Joanna A. Motyl, Joanna B. Strosznajder, Agnieszka Wencel, Robert P. Strosznajder

**Affiliations:** 1Department of Hybrid Microbiosystems Engineering, Nalecz Institute of Biocybernetics and Biomedical Engineering, Polish Academy of Sciences, Ks. Trojdena 4 St., 02-109 Warsaw, Poland; jmotyl@ibib.waw.pl (J.A.M.); awencel@ibib.waw.pl (A.W.); 2Department of Cellular Signalling, Mossakowski Medical Research Institute, Polish Academy of Sciences, 5 Pawinskiego St., 02-106 Warsaw, Poland; jstrosznajder@imdik.pan.pl; 3Laboratory of Preclinical Research and Environmental Agents, Mossakowski Medical Research Institute, Polish Academy of Sciences, 5 Pawinskiego St., 02-106 Warsaw, Poland

**Keywords:** alpha-synuclein, sphingosine-1-phosphate, sphingosine-1-phosphate receptors, sphingosine kinases, Parkinson’s disease

## Abstract

Molecular studies have provided increasing evidence that Parkinson’s disease (PD) is a protein conformational disease, where the spread of alpha-synuclein (ASN) pathology along the neuraxis correlates with clinical disease outcome. Pathogenic forms of ASN evoke oxidative stress (OS), neuroinflammation, and protein alterations in neighboring cells, thereby intensifying ASN toxicity, neurodegeneration, and neuronal death. A number of evidence suggest that homeostasis between bioactive sphingolipids with opposing function—e.g., sphingosine-1-phosphate (S1P) and ceramide—is essential in pro-survival signaling and cell defense against OS. In contrast, imbalance of the “sphingolipid biostat” favoring pro-oxidative/pro-apoptotic ceramide-mediated changes have been indicated in PD and other neurodegenerative disorders. Therefore, we focused on the role of sphingolipid alterations in ASN burden, as well as in a vast range of its neurotoxic effects. Sphingolipid homeostasis is principally directed by sphingosine kinases (SphKs), which synthesize S1P—a potent lipid mediator regulating cell fate and inflammatory response—making SphK/S1P signaling an essential pharmacological target. A growing number of studies have shown that S1P receptor modulators, and agonists are promising protectants in several neurological diseases. This review demonstrates the relationship between ASN toxicity and alteration of SphK-dependent S1P signaling in OS, neuroinflammation, and neuronal death. Moreover, we discuss the S1P receptor-mediated pathways as a novel promising therapeutic approach in PD.

## 1. Introduction

Parkinson’s disease (PD) is a progressive neurodegenerative disorder characterized by both motor and non-motor disturbances. It is currently accepted as a disease of the whole brain, not only the dopaminergic (DA) system. PD follows Alzheimer’s disease (AD) as the most prevalent neurodegenerative disorder worldwide, with an increasing incidence rate. The number of papers investigating the origins of neurodegenerative proteinopathies such as PD and AD has increased in recent years [1,2]. 

The presence of alpha-synuclein (ASN) and amyloid-beta (Aβ) pathogenic forms in both AD and PD, as well as other Lewy Body Disorders (LBDs), partially confirmed the hypothesis of a common neuronal death mechanism [3,4,5,6,7]. Neurotoxicity mediated by pathogenic proteins in parallel with oxidative stress (OS) can augment each other’s effects. ASN is one of the most sensitive and easily available molecules for free radicals as it is present at high levels in the cytosol and presynaptic terminals [8]. Moreover, severe OS is known to inhibit sphingosine kinases (SphKs) and alter sphingolipid homeostasis in favor of ceramide accumulation, which may further intensify the oxidative processes [9]. OS and sphingolipid imbalance may be prominent in the brain, being an organ with a composition high in lipids. Alteration to the “sphingolipid biostat” (see Section 3) is well-documented in clinical and post mortem studies of AD, and also in experimental models [10,11,12,13]. The question of whether bioactive sphingolipid disturbances are crucial for neuronal cell death in PD remains unresolved. However, based on the high susceptibility of DA neurons to oxidative damage, it is a very probable scenario.

The actual therapy of PD and AD is mainly symptomatic, and differs on the basis of clinical symptoms, partially resulting from neurodegeneration preferences for distinct brain regions. However, many neuroprotective approaches to slow down the progression of neuronal death target common neuropathological points such as limiting the toxicity of pathogenic proteins, or the reduction of neuroinflammation and OS, each of which exacerbate the others. Likewise, maintaining bioactive “sphingolipid biostat” may protect against the vicious circle of neurotoxic processes. Sphingolipid homeostasis is principally directed by SphKs, which synthesize sphingosine-1-phosphate (S1P). This potent lipid mediator is an agonist of five S1P receptors (S1P1–5), which regulate cell fate and appear to be a valuable therapeutic target in PD and other neurodegenerative diseases [14,15]. 

In this article, we discuss the recent understanding of SphK/S1P signaling crosstalk with ASN in the context of the PD pathomechanism. This review is divided into chapters, including the role of sphingolipids in ASN toxicity and ASN-related pathological processes such as neuroinflammation. Moreover, we consider S1P-receptor-mediated signaling as a promising therapeutic target in synucleinopathy.

## 2. ASN and Its Role in Physiology and PD Pathogenesis

### 2.1. Physiological Function of ASN

ASN is a small, 140-amino acid protein encoded by the *SNCA* gene. The physiological function of ASN is still under debate [16]. However, due to its presence in synapses, ASN is suggested to be connected with synaptic vesicle regulation and neurotransmission [17,18]. ASN is postulated to be an essential regulator of dopamine (DA) signaling [19]. Moreover, ASN has been reported to modulate axon outgrowth, and was suggested to be involved in compensation mechanisms, implicating corticostriatal glutamatergic plasticity, which occur in early PD [20]. ASN loss within nigrostriatal neurons may lead to their degeneration and neuroinflammation [21]. Distinguishing the physiological or pathological function of ASN may be dependent (at least partially) on its conformational states. The native state of ASN is also under consideration. For many years, ASN has been recognized predominantly as a soluble monomer [22], but further studies revealed that it might also form a folded helical tetramer resistant to aggregation [23,24]. ASN folding may also be a prerequisite for its physiological function in soluble N-ethylmaleimide-sensitive factor attachment protein receptor (SNARE) complex formation in presynaptic terminals and exocytosis. Upon membrane binding, ASN in the form of a cytosolic unfolded monomer changes to a physiologically active multimer, which escorts the SNARE complex [25].

ASN exerts its effect at neuron axons and synapses and acts on diverse cell organelles, including mitochondria and the nucleus, as shown schematically in Figure 1. Over 200 proteins were indicated in an analysis of protein networks in the immediate vicinity of ASN in primary rat cortical neurons [26]. Many of these were derived from the synaptic terminal and vesicle compartments, including those involved in endocytic trafficking. In addition, mRNA-binding proteins and cytoskeletal proteins (included microtubule-associated protein Tau) were identified. The indirect interaction between ASN and axonal microtubules has also been described previously by a mechanism relying on ASN-induced stimulation of protein kinase A-mediated Tau phosphorylation [27]. In the nucleus, ASN may interact with histones, and thus regulate epigenetic control of gene expression [28]. ASN may also be engaged in mitochondrial calcium homeostasis via the promotion of mitochondrial interactions with endoplasmic reticulum (ER). ASN silencing was associated with mitochondrial morphological alterations and decreased calcium uptake [29]. In lipids rafts, which are highly enriched in sphingolipids, extracellular ASN was reported to drive expulsion of the S1P1 receptor, consequently reducing S1P signaling [30]. Moreover, ASN may interact with amyloid-beta precursor protein (APP) and enzymes engaged in APP metabolism, which exists in lipids rafts [31,32,33]. In this way, ASN may exert its toxic effect by enhancing Aβ generation.

### 2.2. The Role of ASN in PD Pathomechanism

In pathological conditions, ASN quickly changes its conformation, becoming prone to aggregation. Smaller forms of ASN such as oligomers and protofibrils gradually change into fibrils and ultimately become deposited into filamentous aggregates known as Lewy bodies (LBs) [34,35,36,37]. Disorders evoked by ASN pathology so-called synucleinopathies are characterized by cytoplasmic ASN inclusions occurring predominantly in neurons, but also in glial cells. Intraneuronal LBs are a feature of LBDs, which, aside from PD, include dementia with LBs (DLB), PD dementia (PDD), and pure autonomic failure (PAF) [16]. Furthermore, glial cytoplasmic inclusions composed mainly from ASN (Papp-Lantos bodies) play a significant role in multiple system atrophy (MSA) [38]. Moreover, ASN pathology also occurs in patients diagnosed with AD in tandem with other diseases. ASN/LBs burden is commonly associated with deposition of Aβ, Tau, transactive response DNA binding protein 43 kDa (TDP-43), or prion protein [7,39]. The accumulation of LBs in the surviving neurons and dystrophic Lewy neurites (LNs) and the degeneration of DA neurons in the *substantia nigra pars compacta* (SNpc) are the main neuropathological hallmarks of PD [37,40]. However, preceding the formation of LBs, ASN oligomers are recognized as a highly neurotoxic form of this protein [36,41]. Similarly to prion-like protein, once misfolded, ASN changes are transduced from one ASN molecule to the other and, by spreading from one cell to its neighbors, the protein pathology can occupy others brain regions [42,43]. 

The cardinal clinical symptoms of PD are related to the impairment of movement, including bradykinesia, muscle rigidity, resting tremor, and postural imbalance [44]. Significantly, over the past few years in PD diagnosis, prodromal non-motor symptoms have captured growing attention, and ASN pathology seems to play an essential role intheir occurrence. These symptoms include hyposmia in very early stages, depression, sleep disorders, and gastrointestinal dysfunctions such as impaired gut motility. These symptoms arise years earlier than classical motor disturbances become clinically evident [45,46]. ASN-immunopositive LBs/LNs are detectable both within the autonomic peripheral and central nervous systems (CNS), and the ASN transmission hypothesis has been confirmed in in vitro/in vivo studies [47]. Moreover, ASN has been detected in biological fluids like cerebrospinal fluid (CSF), plasma, and serum, which partially confirm its extracellular release [48,49]. Furthermore, serum ASN level was found to correlate with a worsening of motor symptoms in the early stages of PD [49]. Six stages of ASN lesions spread, as stated by Braak et al. (2003), describe PD progression, which is in parallel with the appearance of motor and non-motor symptoms. In brief, degeneration starts from the vagus nerve’s dorsal motor nucleus, the olfactory bulb, and the enteric nervous system (ENS) [50,51]. The first symptoms at this stage may be olfaction loss and sleep disturbances, which are recognized as promising diagnostic tools in PD [46]. Simultaneously, pathological changes within the autonomic system—e.g., in the neurons of Meissner’s and Auerbach’s plexus—may result in impaired intestinal motility [51,52]. ASN transmission from gut to brain was also confirmed in an animal model of synucleinopathy [53]. Increasingly, ASN pathology originating within the ENS is recognized as a critical component of the gut–brain axis [52,54]. Recently, an autopsy study of esophageal Lewy body pathology indicated them as a predictive marker of LBDs [55]. 

The classical movement-related PD symptoms appear only a few years or later from the onset of non-motor signs, and the dynamic of ASN pathology spread from nigrostriatal to widespread cortical and brainstem regions responsible for motor function may be crucial [56]. However, the Braak scheme of synucleinopathy is not valid in all types of PD. For example, PD patients may exhibit Lewy pathology in SN, but not in the vagus nerve’s dorsal motor nucleus [57]. In the final stage, ASN pathology gradually affects the cortical areas, including regions controlling higher cognitive processing. ASN-positive cortical LBs have been shown to correlate more highly in terms of both sensitivity and specificity with dementia in PD patients than AD-type pathologies [58]. Interestingly, a greater cortical and limbic LBs burden in DLB than in PDD is an indicator (besides earlier cognitive deficits in DLB) that distinguishes DLB and PDD [59].

Most PD cases (around 90%) are sporadic. However, in the genetic background of familial PD, dominantly inherited ASN gene (*SNCA*) multiplications, as well as several missense mutations (A30P, E46K, H50Q, G51D, A53E, A53T, and A53V), provide evidence for the essential role of ASN in PD [38]. The A53T variant—involving from the substitution of alanine to threonine at position 53—was first described among Italian kindred, as well as three unrelated Greek families who transmitted PD in an autosomal dominant manner [60]. The kinetic features of ASN-A53T protein—i.e., fast nucleation of competent species, continuous seed elongation, and overcoming the surface required for growth—may explain the early onset of PD patients harboring this mutation [61]. However, ASN A53V mutation exhibits a lower tendency to aggregate [62]. Moreover, transgenic mice which overexpressed ASN-A53T developed synucleinopathy and motor impairment, which recapitulates features of human ASN inclusions, and its appearance corresponded with disease onset [63]. Significantly, transgenic mice bearing the A53T mutation developed ENS synucleinopathy and pre-motor symptoms, such as disturbed gut motility [64].

### 2.3. ASN-Protein Network and Its Role in Neurotoxicity

One essential role contributing to ASN toxicity is its interaction with Aβ, which may lead to cross-seeding of aggregation. Almost three decades ago, Uéda et al. recognized an uncharacterized 35-amino acid peptide in sequences of amyloid, which they named the “non-Aβ component” of AD amyloid (NAC) [65]. Its precursor (NACP) was later defined as ASN. The co-occurrence of AD pathology in LBDs and of LBs in autopsied brains of AD individuals is frequent [66]. Aβ has been shown to promote ASN aggregation [67,68]. On the other hand, ASN promotes Aβ secretion and toxicity, probably by the promotion of APP cleavage [32,33]. According to the measured interplay of ASN and Aβ, our last data suggested that exogenous ASN oligomers evoked molecular changes in enzymes involved in APP/Aβ metabolism [31]. In our study, ASN oligomers decreased a disintegrin and metalloprotease domain-containing protein 10 (ADAM10) expression, acting as alpha-secretase [31], which may have enhanced APP metabolism in membrane lipid rafts in the direction of Aβ release. The effect of ASN on APP/Aβ metabolism and other cellular signaling pathways is summarized in Figure 2. 

ASN was suggested to drive Tau pathology, which next to the Aβ burden is a hallmark of AD. In ASN-Tau crosstalk, ASN seems to play a primary role, as its loss in mice reduces the accumulation and spread of Tau, while Tau absence had no significant effect on ASN seeding or spreading [69]. Furthermore, phosphorylated forms of ASN and Tau have demonstrated co-occurrence in the same brain structure, including in the SN and the hippocampus in the 1-methyl-4-phenyl-1,2,3,6-tetrahydropyridine (MPTP) mice model. Furthermore, glycogen synthase kinase 3 beta (GSK-3β) inhibition reduced numbers of p-Tau positive cells, which was accompanied by improved motor performance [70]. 

ASN also modulates mitochondria homeostasis. Disturbed distribution of mitochondria throughout the axon, as well as reductions in axonal mitochondria size, was observed in neurons overexpressing ASN [71]. The most pronounced effect was shown in the A53T variant of ASN, which may suggest a link between genetic background associated with PD and disturbance to mitochondria homeostasis [71].Previously, we showed that silence information regulator 1-sirtuin 1 (Sirt1), a member of the mammalian class III histone deacetylases (HDACs) family, is down-regulated by extracellular ASN [31]. Sirt1 catalyzes deacetylation of peroxisome proliferator-activated receptor gamma-coactivator 1-alpha (PGC-1α), increasing its transcriptional activity, which is a stronger stimulator of mitochondrial biogenesis [72]. Thus, downregulation of Sirt1 triggered by ASN may contribute to the disruption of mitochondrial biogenesis. ASN was shown to alter mitochondria fusion/fission equilibrium, reducing fusion via its unique membrane interaction. Moreover, co-expression of parkin-E3 ubiquitin ligase with ASN was suggested to protect against ASN-induced mitochondria fragmentation [73]. The most recent publication by Wilkaniec et al. (2019) showed that exogenous ASN oligomers evoke parkin S-nitrosylation, which may drive autoubiquitination and degradation of the parkin. This alteration leads to an increase in neuronal cell death, while parkin overexpression works to rescue against ASN toxicity [74]. Parkin seems to be an important regulator of mitochondrial biogenesis, and its significant role in this process appears to be exerted by the aforementioned PGC-1α. Parkin loss in primary skin fibroblasts derived from patients diagnosed with an early onset PD was associated with the blocked action of PGC-1α [75]. ASN localized in the nucleus was suggested to bind to the promoter of PGC-1α under OS conditions, which corresponds with the impaired PGC-1α and its potential downstream genes expression, as well as disturbed mitochondria morphology and function [76]. Furthermore, dynamin-related protein 1 (Drp-1; a fission protein) is recognized as a necessary player in mitochondrial disruption triggered by ASN. Overexpression of ASN leads to mitochondrial fragmentation in a Drp-1-dependent manner [77]. Simultaneously, overexpression of Opa-1 (a fusion protein) was shown to protect cells against mitochondrial fragmentation and ASN-mediated cytotoxicity [77].

Mitochondria dysfunction is an excellent source of reactive oxygen species (ROS), and ASN toxicity is firmly attributed to oxidative/nitrosative stress (OS/NS). ASN may drive its toxic effect by the stimulation of nitric oxide (NO) production. Exogenous ASN was reported to activate nitric oxide synthase (NOS) in immortalized mouse hippocampal cells [78] and striatal cytosolic NOS in rat brain, which may be implicated in a mechanism for reduced DA transporter activity [79]. Sphingolipid homeostasis imbalance then plays a role in this vicious circle. OS is an essential switch of SphK1 activity, which is inhibited under oxidative injury [9]. Simultaneously, ASN can hinder SphK/S1P signaling, which was shown to enhance OS [31,80]. ASN impact on Sphk/S1P metabolism and action is presented in Figure 3 in more detail. For short, ASN may affect S1P1 effector proteins, such as pro-survival serine/threonine protein kinase B (PKB), also known as Akt kinase and brain-derived neurotrophic factor (BDNF). Chung et al. (2011) showed that, in physiological conditions, ASN could be a co-regulator of Akt activation as stimulated by growth factors. However, its mutation or overexpression disturbed the process, which was confirmed by the inhibition of insulin-like growth factor 1 (IGF-1)-induced Akt activation [81]. Reduced phosphorylation/activation of Akt kinase was also observed in ours’ and others’ studies in neuronal cells treated with exogenous ASN [31,82]. ASN was also shown to inhibit BDNF signaling, leading to DA cell death [83]. 

ASN is involved in the epigenetic modification of gene expression, and in this way may also counteract S1P signaling. In experimental models of PD, it has been suggested that ASN translocates to the nucleus and exerts toxicity upon neurons via its interaction with histones [91]. ASN was shown to inhibit histone H3 acetylation processes by direct association with the histones [28]. It can therefore be hypothesized that ASN may disturb the action of histone acetyltransferases (HATs). Moreover, its nuclear targeting is associated with the A53T and A30P mutations found in familial PD [28]. HATs oppose the enzymatic activity of HDACs, i.e., a class of enzymes with a repressive effect on histone acetylation. As a result, some HDACs inhibitors may exert a neuroprotective effect in experimental PD models [92]. In contrast to ASN, nuclear S1P can inhibit histone deacetylases HDAC1 and HDAC2, preventing deacetylation of histone H3 [90].

ASN toxicity may also be attributed to the over-activation of DNA repair enzyme poly(adenosine 5′-diphosphate–ribose) polymerase-1 (PARP-1), followed by caspase-independent but PARP-1 dependent apoptosis, called “Parthanatos.” Intrastriatal injection of preformed ASN fibrils was shown to induce PAR accumulation, DA neurons damage, and motor disturbances in mice [93]. Further, genetic deletion of PARP-1 (or its pharmacological inhibition) diminished ASN toxicity and loss of DA neurons, and also improved behavioral performance [93]. In another study, PARP-1 inhibition by Veliparib was found to attenuate ASN accumulation, protect DA neurons, and improve the motor ability of transgenic mice carrying the A53T mutation [94]. An important event connected with PARP-1 is its role in promoting nuclear factor kappa B (NF-κB) activation, as well as downstream pro-inflammatory mediator expression in microglia [95]. Similarly, the recruitment of NF-κB and activator protein 1 (AP-1) transcription factors was observed under A53T-mediated microglial activation [96].

## 3. S1P Metabolism and Its Role in PD

S1P is recognized as the most potent bioactive sphingolipid, regulating proliferation, growth, and cell survival [15,97,98,99]. Its effect is contrary to the apoptosis and growth arrest mediated by ceramide, which accumulates in pathological conditions [100]. The dynamic balance between S1P and ceramide is called the “sphingolipid biostat,” and the key enzymes for the regulation of this equilibrium are sphingosine kinases (SphKs). SphK1 and SphK2 isoforms are encoded by genes located on chromosome 17 (17q25.2) and 19 (19q13.2), respectively. Furthermore, as a result of alternate splicing, there are differences within each isoform sequence, giving rise to several variants of SphK1 and SphK2 [101,102]. With high similarity in about 80% of their amino acid sequences, the central and N-terminal regions distinguish the individual kinases, determining their subcellular localization and biochemical properties [103]. As a result, the S1P pools synthesized by SphK1 and SphK2 show differences in their impact on the cell [104]. In the brain, among SphK1 activators, one essential role seems to be the promotion of nerve growth factor (NGF), markedly enhancing not only its enzymatic activity and plasma membrane translocation but also gene expression [105,106]. Simultaneously, SphK1/S1P signaling was reported to stimulate transcription of glial cell line-derived neurotrophic factor (GDNF)-induced growth-associated protein 43 (GAP43), being a crucial axonal component [107]. In contrast to SphK1, a mainly cytosolic enzyme, SphK2 has many cell locations which vary depending on the cell type. In many cell lines, SphK2 was found essentially in the nucleus, but in others it was detected in plasma membranes, ER, Golgi, mitochondria, and the cytosol [108]. In contrast to SphK1, SphK2 was first recognized predominantly as a pro-apoptotic protein, and its properties resulted from its putative BH3 domain and induction of the mitochondrial apoptotic pathway [109]. The above and many other studies have shown that the S1P pool synthesized by this isoform may act independently of S1P receptors. For example, S1P synthesized by mitochondrial Sphk2 participates in BAK-pro-apoptotic Bcl-2 member activation and hence outer-membrane permeabilization and cytochrome c release [110]. Likewise, nuclear SphK2 could be associated with DNA synthesis inhibition and cell cycle arrest at the G1/S phase checkpoint [111]. Moreover, S1P may directly bind to, and enhance, the proteolytic activity of β-site APP cleaving enzyme 1 (BACE1), thereby promoting Aβ generation [112]. However, Sphk2/S1P signaling may be vital for epigenetic gene expression regulation, and mitochondria homeostasis beyond its pro-apoptotic effect [86,87,90]. Regardless of the diversity, SphKs may compensate for each other in their fundamental function—i.e., S1P synthesis. Deletion of both SphK isoforms resulted in undetectable levels of S1P in mice, leading to embryonic lethality [113]. In contrast, deletion of either SphK1 or SphK2 alone resulted in mice which were viable, fertile, and had no apparent abnormalities relative to their wild-type counterparts [113,114].

S1P level is regulated also by S1P-degrading enzymes, such as S1P phosphatases (SPPs) that convert S1P to sphingosine (Sph) and S1P lyase (SPL), which terminally cleaves this sphingolipid to hexadecenal and ethanolamine phosphate [115]. Nowadays, compounds that mimic S1P action are recognized as promising in studies concerning a subset of neuronal diseases [116,117,118]. On the other hand, SPL-encoding gene mutation leading to the inborn error, referred to as sphingosine phosphate lyase insufficiency syndrome (SPLIS), could be toxic to neurons [119].

Once synthesized, S1P can either transduce its signal inside the cell, or it can be transported extracellularly and act via its receptors in an autocrine (i.e., “inside-out signaling”) or paracrine manner [120,121]. Transport of this bioactive sphingolipid is carried out by spinster protein 2 (Spns2) and the ATP-binding cassette (ABC) family [122,123]. Spns2 is also implicated in transport of phosphorylated fingolimod (FTY720), which is S1P analogue used in the clinic [124]. S1P is a ligand of five S1P-specific GTP-binding protein (G-protein)-coupled receptors (GPCRs), called S1P1-5. In this way, S1P may play a fundamental role within the cell, ranging from neuro- and angiogenesis [113] to the regulation of crucial function in the mature organism, such as the immune response [125,126]. 

S1P also plays an important role within brain processes, such as synaptic transmission and memory formation [97]. Four of the five members of S1P receptors (S1P1, S1P2, S1P3, and S1P5) are expressed in hippocampal neural progenitor cells [127]. These four S1P receptors are located both in neurons as well as glial cells, and their expression may differ during all stages of brain maturation [118,128,129]. Their expression pattern may also dynamically change depending on the cell activation state. For example, in resting microglia, the expression of S1P1 and S1P3 were expressed at a high level but were down-regulated after lipopolysaccharide (LPS) challenges [130]. Conversely, in reactive astrocytes of multiple sclerosis (MS) lesions and primary human astrocytes, as well as in cell lines cultured with proinflammatory cytokines, S1P1 and S1P3 receptors were shown to be up-regulated [131]. 

S1P1 is the most abundant S1P receptor within the CNS. Mice with a S1P1 receptor deletion are characterized by abnormal formation of the neural tube and blood vessel failure which leads to embryonic death. Mortality at the embryonic stage is not observed upon deletion of the other four S1P receptors [113]. Another study also revealed the emerging role of S1P1 in neurogenesis and S1P1-S1P3 in angiogenesis [132]. In the mature CNS, S1P regulates among others the activation of neuronal progenitor cells and their migration to affected brain and spinal cord regions [133,134]. 

Our current understanding of the role of SphKs/S1P signaling in PD is scant, and has been gleaned mainly through the use of experimental models, such as 1-methyl-4-phenylpyridinium (MPP+)/MPTP. Our previous studies have shown decreased expression and activity for SphK1 in SH-SY5Y cells, subjected to DA neurotoxin MPP+, as well as in the whole midbrain of C57BL/6 mice administered with MPTP [135,136]. Recently, in the same MPP+ in vitro model, gene expression and protein levels of SphK1 were also shown to be down-regulated [137]. Additionally, inhibition of SphK1, by miRNA-targeted SphK1 potentiates its downregulation mediated by MPP+, which appears in parallel with the aggravation of neurotoxic effects [137]. Furthermore, SphK1 was shown to be downregulated in the striatum of MPTP-administered mice [138]. Similarly, a decrease in mRNA level of both SphK1 and SphK2 was found within SN of mice exposed to MPTP, as well as in MN9D cell line forced with MPP+ [86,139]. 

We have also shown that pharmacological inhibition of SphKs exerts a pro-apoptotic effect on DA cells, while S1P and its receptor modulator—FTY720—may offer neuroprotection in the MPP+/MPTP model [135,136,140]. Further supporting this conclusion, the inhibition of miRNA-targeted SphK1 resulted in SphK1 overexpression, protecting SH-SY5Y cells against MPP+ neurotoxicity [137]. Additionally, an SphK1 activator (2-Hexyl-N-[2-hydroxy-1-(hydroxymethyl)ethyl]-3-oxo-decanamide (K6PC-5)) prevented MPP+-induced alterations in the expression of the pro- and anti-apoptotic Bcl-2 protein [141]. 

It has been shown that the level of ceramide—i.e., a sphingolipid with opposing function to that of S1P—was altered in the bodily fluids of PD patients. Several ceramides and their derivative species were reported to be increased in the plasma of PD subjects as compared to control individuals, and were highest among patients diagnosed with cognitive impairment [142]. Furthermore, in PDD, the elevation in several plasma ceramide species was negatively correlated with verbal memory capacity or linked with higher incidences of some neuropsychiatric signs, such as anxiety, hallucinations, and sleep disturbances [143]. Ceramide metabolism may also be altered by the absence of Leucine-rich repeat kinase 2 (LRRK2), mutation of which causes inherited PD. Concentrations of ceramide 18:1 in mouse brain extracted lipids was detected to be significantly elevated in LRRK2^−/^^−^ mice as compared to the wild-type counterparts [144].

## 4. The Relationship between ASN Toxicity and Alteration of S1P and Ceramide Signaling

The toxicity of ASN is understood to develop by the loss of its essential physiological functions, and a switch to a role in explicitly pathological processes. The harmful effect of ASN in the cellular model is most often conferred by extracellular treatment with diverse ASN species, or it may result from genetic modifications of ASN leading to its overexpression [145]. Animals carrying ASN point mutations—e.g., A53T, among others—are used as in vivo models of synucleinopathy [63]. The majority of reports regarding sphingolipid connection with ASN focus on ceramide [146,147]. 

Much less is known about the role of SphKs/S1P signaling in ASN toxicity regulation. Indirectly, a few studies indicate a significant mutual impact of SphK1/S1P signaling on ASN turnover and toxicity [31,80,140]. The effects of ASN and S1P signaling are summarized in Figure 3. Over the past decade, much attention has been paid to the role of S1P receptor modulators in ASN-induced toxicity, and most of these studies have shown their beneficial effect [64,148,149].

A few studies reported that SphK1/S1P may affect ASN expression, its release outside the cell, and its biological function. Simultaneously, ASN may influence the SphK1/S1P signaling as well. Our previous results suggested that inhibition of SphK induces ASN release into the extracellular space in SH-SY5Y cells, which overexpress ASN [140]. Moreover, pharmacological inhibition of SphKs was found to increase the secretion of ASN from synaptoneurosomes to extracellular spaces, and potentiate its liberation as mediated by OS. Simultaneously, extracellular ASN decreased SphK1 expression but not that of SphK2 [85]. Recently, we demonstrated that exogenous ASN oligomers reduced the activity and protein level of SphK1, leading to molecular alterations and cell death [31]. Other studies have indicated that pro-apoptotic conditions, evoked by a high concentration of ASN, or prolonged time of exposure, decreased the expression level of SphK1, yet upregulated the expression of the SphK2 isoform [80].

SphK1 may also be implicated in ASN post-translational modification. Screening of siRNA libraries implicated the role of the SphK1 gene in the possible phosphorylation of ASN on Ser129 [150]. ASN phosphorylated on this residue occurs in synucleinopathy lesions [151], but its requisite role in the ASN aggregation process is still debated and may even be a signal for degradation [152,153].

Extracellular ASN may affect also S1P receptor signaling. Zhang et al. (2017) have reported that both wild-type recombinant ASN and its PD-linked mutated form (A53T) drive selective S1P1 receptor-mediated signaling uncoupling from G inhibitory (Gi) protein in SH-SY5Y cells [80]. This disturbance of S1P1-dependent signaling contributes to ASN-induced impairment of platelet-derived growth factor (PDGF)-mediated chemotaxis. It is worth mentioning that a stronger effect was exerted by the A53T form, confirming the important role of ASN mutation in its toxicity. Interestingly, within two major S1P receptors expressed in SH-SY5Y cells (S1P1 and S1P2), ASN only exerted an effect on S1P1 uncoupling, confirming the crucial role of S1P1 in ASN-mediated aberrant cell motility [80]. Furthermore, extracellular ASN was found to mediate the expulsion of S1P1 from lipid rafts, leading to the subsequent suppression of Gi signals in SH-SY5Y cells lipids rafts [30]. 

Moreover, ceramide was shown to be engaged in ASN-induced neuronal insults. ASN overexpression in human medulloblastoma cells leads to ceramide accumulation [154]. Myriocin, an inhibitor of serine palmitoyl transferase (SPT) that decreases de novo ceramide synthesis, mitigates ASN-mediated toxicity in Drosophila eyes [154]. Several genes implicated in LBDs pathology have been postulated to affect ceramide metabolism [155]. One of these could be the ASN gene (*SNCA*), although its impact on ceramide metabolism could be indirect. *SNCA* deficiency in mice was shown to decrease brain palmitate uptake [84], which may suggest that ASN deficiency affects de novo synthesis of ceramide by reducing the availability of its substrate. 

Toxicity mediated by ASN may be related to the dysfunction of lysosomal glucocerebrosidase 1 (GCase1), converting glucosylceramide (GlcCer) to ceramide and glucose. Mutation in gene-encoding GCase1 is a genetic risk factor of PD and DLB [156,157]. Taguchi et al. (2017) reported that accumulation of GlcCer, glucosylsphingosine (GlcSph), Sph, and S1P accelerates ASN aggregation in vitro. However, only GlcSph and Sph-induced ASN species promoted endogenous ASN aggregation in neuronal cells [158]. GCase1 deficiency, induced by its gene mutation, exacerbated the ASN oligomerization induced by MPTP administration [159]. Furthermore, GCase1 loss, which resulted in higher vulnerability to MPTP intoxication, was shown to be related to the accumulation of ASN, as its knockout significantly improved MPTP-induced DA cell loss and motor deficits [159]. 

Moreover, some form of antagonism exists between ASN and S1P impact on S1P receptor downstream proteins (Figure 3). For example, ASN may block neurotrophic and pro-survival activities of proteins essential in S1P1 action, including the BDNF/Akt kinase signaling pathway [81,83,88,89]. The differing effects of ASN and S1P are also exerted on mitochondrial homeostasis. As mentioned before, PGC-1α action is inhibited by ASN [76]. On the other hand, PGC-1α seems to be an important downstream protein in S1P1 receptor-mediated neuroprotection. Mitochondrial SphK2/S1P signaling has been suggested to protect DA neurons in an MPTP/MPP+ model, partially by way of PGC-1α activity [86]. Moreover, S1P produced by SphK2 in mitochondria via its interaction with prohibitin 2 (PHB2) regulates cytochrome c oxidase assembly and respiration [87]. Similarly, SphK1 was shown to maintain mitochondrial homeostasis. One study in the C. elegans model found that SphK1 recruitment to the mitochondria is required at an early step for the activation of the mitochondrial unfolded protein response (UPR^mt^) under neuroendocrine stress signaling [160]. S1P and ASN also vary in epigenetic regulation of gene expression. In contrast to ASN, which hinders histone H3 acetylation, nuclear S1P can inhibit histone deacetylases HDAC1 and HDAC2, preventing deacetylation of histone H3 [90].

## 5. The Roles of Sphingolipid Signaling and ASN in Neuroinflammation

Neuroinflammation is inseparably bound to the pathogenesis of PD [161,162]. However, it remains controversial as to whether it is a primary or secondary event to co-existing neurodegeneration, and whether other pathological processes like proteins aggregation, OS or mitochondria failure are necessary triggers. Lewy pathology inclusions which stained positively for ASN were discovered in embryonic DA neurons implanted into the striatum of patients with advanced PD [163]. Further temporal analysis of fetal tissue graft deposits derived from patients who died in the period from 1.5 to 16 years after transplantation has shown the emergence of microglial activation in all examined cases, while ASN pathology affected only the older grafts [164]. 

These observations suggest that neuroinflammation may far precede the appearance of ASN pathology. Another study found that activation of inflammasome—an important component of innate immunity—leads to ASN aggregation [165]. Likewise, ASN is a potent inducer of the inflammatory response. Overexpression of its mutated form in transgenic mice in combination with LPS may be used to investigate the mechanisms of the ASN and neuroinflammation double-hit that lead to chronic PD neurodegeneration [166]. Moreover, ASN was shown to drive a strong cytokine response in peripheral blood mononuclear cells from PD patients and their age/gender-matched controls [167]. On the other hand, the loss or dramatic reduction of ASN level, similarly to its overexpression within the brain, results in neuroinflammation and DA cell death, which confirm the vital role of ASN under physiological conditions [21]. 

S1P is also an important regulator of the immune response. A growing body of evidence suggests that S1P signaling modulation is a potential therapeutic target in MS and other diseases involving chronic inflammation [125,126]. S1P concentration in the CSF of early stage MS patients was elevated compared to the control individuals [168]. The S1P gradient plays an important role in lymphocyte trafficking [169]. The non-selective S1P receptor modulator FTY720, in its active phosphorylated form, binds with high affinity to lymphocytic S1P1, leading to its internalization and degradation. This reduces lymphocyte exit from the lymph nodes, limiting recirculation of autoreactive T cells affecting the CNS, which at least partially explain the therapeutic effect of FTY720 in MS [170]. On the other hand, analysis of lymphoid tissue treated with an inhibitor of SPL revealed the relationship between SPL decreased activity/S1P accumulation and reduced numbers of circulating lymphocytes [171]. 

Moreover, S1P synthesizing enzyme SphK1 may influence neuroinflammation by controlling the activity of glial cells. SphK1 is one of the genes up-regulated in A2 astrocytes [172]. Importantly, SphK1 expression was down-regulated by MPTP neurotoxin, inducing a switch from alternative A2 to classical A1 phenotype [138]. SphK1 is also suggested to be a novel marker of the microglia with immunomodulatory M2b phenotype [173]. The primary microglial cultures challenged with LPS exhibit an initial tendency to increase SphK1, followed by significant down-regulation after a more extended exposition for the toxin [173].

FTY720’s therapeutic effect was confirmed in several PD models induced by MPTP, which mimics neuroinflammatory changes within the brain [89,135,174]. One of the proposed mechanisms of FTY720 and SEW2871—a selective agonist of S1P1—action in MPTP mice brain was to prevent neuroinflammation signs such as astrogliosis and microgliosis in SNpc and suppression of pro-inflammatory markers like tumour necrosis factor-alpha (TNF-α) cytokine and glial fibrillary acidic protein (GFAP), as well as elevating BDNF level in the striatum. Importantly, SEW2871 was shown to have a comparable neuroprotective effect to FTY720, which raises the possibility that the anti-inflammatory effect of these compounds could be mainly dependent on S1P1 receptor activation [89]. Similarly, Yao et al. (2019) have revealed that FTY720 suppresses microglial activation in the SNpc, attenuating motor dysfunction and the loss of DA neurons in MPTP-forced mice. Furthermore, FTY720 pre-treatment reduced the production of crucial pro-inflammatory cytokines (interleukin (IL)-6, IL-1β, and TNF-α) in BV-2 microglial cell line treated with MPP+ [174]. Moreover, FTY720 decreased the apoptosis of SH-SY5Y cells conditioned with medium from MPP+-treated BV-2 cells. Finally, this molecule reduced ROS generation and p65 phosphorylation, and affected the activation of the NLR family pyrin domain containing 3 (NLRP3) inflammasome and caspase-1 in primary microglia and BV-2 cells [174].

## 6. Pharmacological Targets Implicating S1P Signaling in PD: The Role of FTY720

The disturbed “sphingolipid biostat” seems to be an important element of the neurodegeneration cascade. It has been shown to be significantly affected in human autopsy brains and experimental studies on neurological disorders such AD, PD, and Huntington’s disease (HD), making it a potential target of therapy [15,118,175,176]. However, among the aforementioned diseases, PD is still the most enigmatic in this field. The modulation of sphingolipids balance in favor of S1P is suggested to be a promising target for neuroprotection (Figure 4). This could potentially be achieved by decreased activity of the SPT—the rate-limiting enzyme in de novo ceramide synthesis or SPL inhibition—which terminally cleaves S1P. Enhanced S1P signaling could also be reached by the activation of SphK1. In line with the above, several studies have indicated neuroprotective properties of SphK1 activator K6PC-5 in MPP+ [141], in OGDR-induced stress [177], and in cellular models of HD [178]. Moreover, SphK1 overexpression mitigated the MPP+ toxic effect in SH-SY5Y cells [137]. However, modulation of SphK activity or expression would need to be carried out with extreme caution, and until now has only been considered for anti-cancer therapy, where the target is SphK/S1P lowering [179]. In the nervous system, S1P concentration should always be carefully regulated and maintained at the appropriate level, as S1P overabundance could be as harmful as its deficiency. For example: SPL dysfunction, leading to S1P accumulation and reduced levels of phosphatidylethanolamine arising from the products of S1P degradation, may disturb the autophagy of pathological proteins such as ASN and APP [180].

Another toxic effect elicited by Sphk1 inhibition and “sphingolipid biostat” disturbances is oxidative stress. Therefore, anti-oxidative agents, like Sirt1 activators, may be significantly promising in PD therapy, since Sirt1 may also reduce ASN accumulation by regulating its autophagy [181,182].

Direct S1P receptor activation seems to be associated with a lower risk of side effects than the regulation of the S1P metabolism, and hence S1P level itself. Thus, the S1P receptors are a much better target, especially since their modulators are already used in the clinic. Moreover, by exploiting a common mechanism underlying neurodegenerative diseases, these modulators are ideal candidates for a drug repositioning strategy. So far, three have been approved by both the US Food and Drug Administration (FDA) and the European Medicines Agency (EMA) in MS therapy. 

In 2010, FTY720 (trade name Gilenya) was given approval for treatment of the most common course of MS, called “relapsing-remitting” (RRMS). This represents the first oral disease-modifying therapeutic approved for MS [170]. Importantly, siponimod (also known as BAF312, trade name Mayzent), and most recently, ozanimod (trade name Zeposia), which selectively target S1P1 and S1P5, have been approved also for patients with secondary progressive MS, which is heavily challenging from a therapeutical perspective [183,184]. FTY720 in its active, phosphorylated form is a nonselective modulator of S1P receptors (except S1P2). The phosphorylation of this pro-drug is catalyzed with the greatest efficiency by SphK2 [185]. The drug has a predictable pharmacokinetic profile that provides sufficient once-daily oral dosing. FTY720 has a long half-life of 6–9 days, and it takes 1–2 months to reach steady-state pharmacokinetics after daily dosing [186]. However, a common adverse event during FTY720 therapy is progression from being clinically asymptomatic to exhibiting grade 4 lymphopenia [187,188]. Potential targets for the action of FTY720 are summarized in Figure 5. 

Besides its well-known mechanisms, which operate by sequestration of autoreactive lymphocytes within the lymph nodes and limiting of their migration into the CNS, FTY720 exerts a direct effect on the S1P receptors expressed on neuronal and glial cells [189]. FTY720 has also been reported to enhance the properties of the blood–nerve barrier [190,191]. The mechanism of FTY720 action is both S1P-receptor-dependent, as well as mimicking the intracellular action of S1P. One of the crucial findings in this field was that FTY720 enters nucleus and, after SphK2-mediate phosphorylation, inhibits HDAC class I [192]. It was shown to inhibit hippocampal HDAC enzymatic activity in severely immunodeficient mice and to enhance acetylation of histone lysine residues connected with the epigenetic regulation of learning and memory genes. Thus, FTY720 facilitated the extinction of fear memory in mice in a mechanism independent of its well-established immunosuppressive activity [192]. Another study indicated the protective effect of FTY720 in ischemia-induced neuroinflammation and brain injury, which was associated with microglial switching from the classical M1 to the alternative M2 phenotype. The authors also reported that nuclear SphK2 activity, inhibition of HDAC1, and suppression of deacetylation of key anti-inflammation transcriptional regulators were essential in this process [193].

FTY720 may also exert its neuroprotection by pro-survival kinases downstream of S1P receptors. One of these is phosphatidylinositol 3-kinase (PI3K)-Akt kinase, signaling of which is disturbed in PD, while its activation represents an important target for neuroprotective therapy [194]. One of the well-known activators of the PI3K-Akt signaling pathway is BDNF [195]. This trophic factor is recognized to be crucial in DA cell survival. It reaches high expression is SNpc, but is significantly reduced in PD patients, while its recovery is an important target for neuroprotection [196,197]. Moreover, ASN over-expression was shown to reduce BDNF expression [198]. The Akt-dependent signal is downstream—at least in part—of S1P1 [199] and FTY720, which may stimulate expression of both Akt and BDNF. FTY720 was reported to be involved in Akt activation in in vivo PD models [135,200]. Moreover, Ren et al. (2017) found that FTY720 may activate extracellular signal-regulated kinases (ERK1/2), which, like Akt, can increase the expression of BDNF [200]. 

In recent years, growing attention has been focused on the role of FTY720 in synucleinopathies. Many of these studies indicate that the therapeutic effect of FTY720 is mainly based on the up-regulation of the BDNF expression. In genetic PD models, represented by transgenic mice with *SNCA* mutation A53T, FTY720 administration decreased ASN pathology in the ENS, reduced constipation, and improved gut function, which all seem to be related to BDNF stimulation [64]. In another genetic PD model induced by ganglioside GM2 synthase mutation, FTY720 reduced brachial plexus synucleinopathy, increased BDNF level, and improved PD-associated age-onset motor symptoms and bladder function [148]. It is likely that this drug may act in the early stages of progressive synucleinopathy, while an ameliorating effect on both motor and non-motor symptoms is still possible. 

FTY720 also acts on BDNF in an MSA-like ASN pathology exhibited by oligodendrocyte cell lines. Drug restored reduced by ASN overabundance BDNF expression via histone 3 acetylation on BDNF promoter 1 [149]. BDNF up-regulation may result from S1P1 receptor activation. In a study by Pépin et al. (2020), both FTY720 and the selective S1P1 agonist SEW2871 abolished the lowering of BDNF protein level induced by MPTP [89].

Consistent with these findings, the neuroprotective effect of FTY720 has also been shown in PD models induced by 6-hydroxy-dopamine (6-OHDA) and rotenone [200,201], as well as in acute and subacute MPTP challenge [89,135,174]. FTY720 was reported to activate some pro-survival pathways, influencing DA neuron protection, which coincided with improved locomotor activities. In contrast, in another study of subacute MPTP administration to both wild-type and transgenic mice with ASN mutation A30P, FTY720 had no neuroprotective effect on DA neurons [202]. Further data are needed to confirm or deny the neuroprotective effects of FTY720 in PD. 

To date, FTY720 has been shown to exert a neuroprotective influence in both genetic synucleinopathy and in some toxicological PD models, but it fails in the mixed PD paradigm, i.e., mice with genetic synucleinopathy treated with MPTP toxin.

An important discovery from recent years has been the new generation of compounds based on the structure of FTY720. Modification with moieties based on ceramide C2 (FTY720-C2) and a β-triphenylphosphoniumpropanamide (FTY720-Mitoxy) were shown to determine compound potency and mitochondrial localization, respectively. Both FTY720 analogues penetrate the blood–brain barrier, and unlike the FTY720, are not metabolized into phosphorylated species that are responsible for S1P1-dependent lymphopenia and immunosuppression [203,204]. All FTY720s were reported to increase BDNF expression and stimulate the catalytic activity of protein phosphatase 2A (PP2A), function of which is disturbed in synucleinopathy. Finally, all FTY720s protected MN9D cells in TNF-α-mediated cell death [205]. FTY720-Mitoxy was also shown to increase the levels of BDNF mRNA, as well as that of other growth factors such as GDNF and NGF, in an oligodendrocyte cell line, while FTY720 and FTY720-C2 increased levels of NGF only [206]. Subsequently, it was found that FTY720 and its derivatives have a protective mechanism of action through their impact on neuronal microRNAs. For example, C2-FTY720 up-regulated miR-7a-5p, which may lead to reduced ASN protein level [207]. Likewise, in mice expressing human ASN in oligodendrocytes, FTY720-Mitoxy was demonstrated to increase GDNF and decrease brain miRNAs that reduce GDNF expression [208]. 

FTY720 was also reported to improve cognitive as well as some motor impairment, in harmaline-induced rats model of essential tremor [209]. FTY720 may also offer protection in a model of neurotoxicity induced by human prion protein [210], which, in the light of the ASN pathogenic spread hypothesis, supports the FTY720 significance in synucleinopathies.

## 7. Therapy Focused on Lowering ASN Levels

ASN has the potential to be a leading molecular target in PD therapy (and that of other synucleinopathies), although many questions arise around its pathology, which make realistic cure remain distant. Briefly, ASN-based management is focused on reducing the number of intracellular ASN aggregates, inhibiting seeding and spread from cell to cell, controlling release to the extracellular space, and promoting degradation [211]. Another therapeutic possibility relies on lowering ASN synthesis, which arises partially from the observation that multiplications in *SNCA* genes are relatively common in familial PD cases [212]. Accordingly, several lines of evidence have shown that moderate lowering of ASN level in rodents and primates does not evoke any adverse effects, and may even mediate neuroprotection in toxicological models [213,214]. In contrast, it has also been reported that ASN knockdown in DA neurons of primates evoked nigrostriatal degeneration [215]. Additionally, the intensive studies of the Südhof group underline the role of ASN in synaptic plasticity, in long-term regulation of presynaptic function, and in supporting nerve terminals against injury [25,216]. Their data also emphasize some problems in studying the role of ASN affected by the presence of β-synuclein in the nerve terminals of the brain, as well as the high degree of co-expression of α- and β-synuclein, demonstrating the potential for functional redundancy [217]. Fortunately, the involvement of γ-synuclein can be eliminated as it is abundant mainly in some neurons of dorsal root ganglia, and mainly only in the peripheral nerves. In primary human cortical astrocytes, γ-synuclein was reported to increase their proliferation, followed by enhanced expression of BDNF [218]. The above findings should be taken into account when considering the application of therapeutic strategies leading to elimination of ASN.

## 8. Conclusions

This review has demonstrated an important relationship between bioactive sphingolipids and ASN protein, which easily changes conformation state, and is involved in the pathogenesis of PD. One of the remaining questions surrounds the role of ASN in synaptic endings. In PD pathology, ASN very easily interacts with Aβ, Tau, and other proteins, and it may induce changes in their conformation, thereby increasing their toxicity, and promoting SNARE complex formation. Through its interaction with Aβ, ASN may lead to irreversible alterations in early stages of pathology, and to mitoptosis, synaptosis, and neuronal death. When present in synapses in high amounts, ASN modulates the expression and activity of Sirt1 as well as enzymes involved in APP metabolism and Aβ level, and contributes to disturbances in the “sphingolipid biostat.” Through activation of OS and inhibition of SphK1, ASN may likely decrease S1P concentration with a concomitant up-regulation of ceramide level. All of these events may be responsible for a molecular “vicious circle” and synaptosis, which is the first and most important event in PD. However, the mechanism(s) of ASN-dependent synaptic and mitochondrial dysfunctions in PD are not fully understood. The interactions of ASN with sphingolipids in alterations of protein translocases of the outer and inner mitochondria membranes (TOM and TIM) should be elucidated in further research. ASN, by its interaction with lipids, affects not only mitochondrial transport but also microtubule–kinesin interaction, which may influence axonal transport. Little is known about ASN interaction with ceramide in mitochondria. The influence of ASN on cardiolipin level and mitochondria respiratory supercomplex function and permeabilization was described previously, but not its interaction with bioactive sphingolipids. The significance of ASN and its crosstalk with sphingolipids in other subcellular organelles, such as the lysosome, and also its significance in lysosome-mediated mitophagy, deserve further study. Furthermore, the question of whether ASN is significantly engaged in the alteration of mitochondria-ER communication as well as in mitochondrial dysfunction and fragmentation in PD remains to be elucidated. 

Other open questions concern the role of ASN interaction with bioactive sphingolipids in plasma membrane lipid rafts. Their crosstalk may alter a membrane’s physio-chemical properties, leading to the formation of pores and then to the secretion of ASN into extracellular compartments, and transduction of pathological protein alterations to neighboring cells. The roles of particular bioactive sphingolipids in aggregation, fibrilization, oligomerization, and interaction with ASN are still to be fully explained. Finally, it is unclear whether alterations of bioactive sphingolipid homeostasis and S1P receptor signaling pathways play a crucial role in the pathogenesis of PD, and if this discovery will open up new pathways in therapy of these diseases. The important question is whether the use of S1P receptor modulators such as FTY720 or siponimod, or specific agonists of S1P receptors (S1P1 or S1P3), will improve prospects in therapy of PD. Further studies are warranted to address these issues.

## Figures and Tables

**Figure 1 ijms-22-06277-f001:**
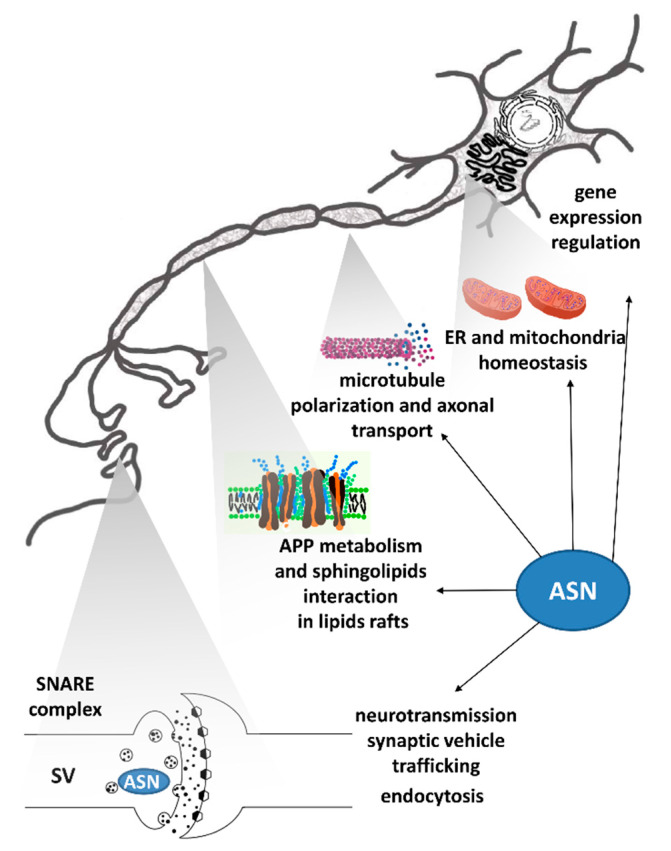
ASN function within different cellular compartments. Abbreviations: ASN—alpha-synuclein; APP—amyloid-beta precursor protein; ER—endoplasmic reticulum; SV—synaptic vesicles; SNARE complex–soluble N-ethylmaleimide-sensitive factor attachment protein receptor complex.

**Figure 2 ijms-22-06277-f002:**
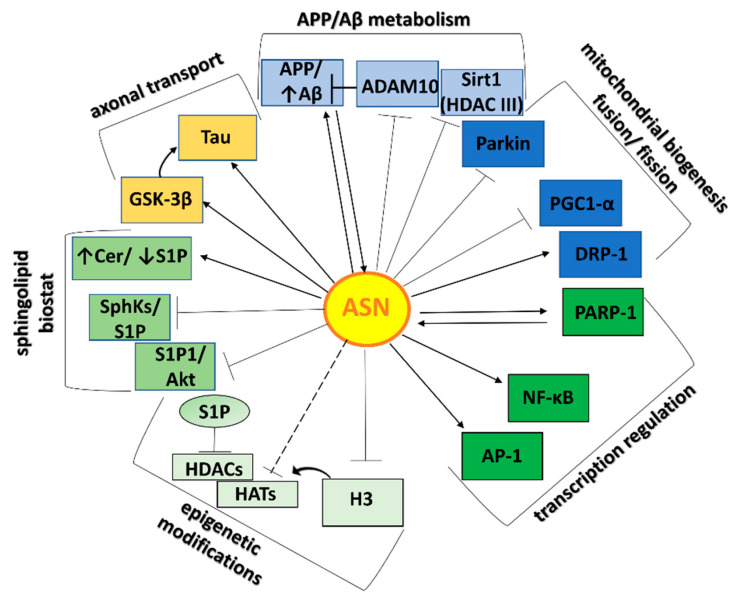
Toxic ASN-protein network and its effect on biological processes. The influence of ASN on individual proteins is marked with arrows that terminate with → (activation), or ┴ (inhibition/deterioration). A dashed line between ASN and HATs and a little arrow from H3 to HATs/HDACs mean that ASN inhibits histone acetylation by directly associating with histone H3, which masks residues required for acetylation [28]. Therefore indirectly, ASN can perturb the HATs/HDACs balance on the side of acetylation inhibition. Proteins interacting with ASNs have been grouped and color coded into categories depending on their biological function. Abbreviations: ASN—alpha-synuclein; APP/Aβ—amyloid-beta precursor protein/amyloid-beta; Sirt1 (HDAC III)—silence information regulator 1-sirtuin 1 (histone deacetylases class III); ADAM10—a disintegrin and metalloprotease domain-containing protein 10, acting as alpha-secretase; parkin—E3 ubiquitin ligase; PGC-1α—peroxisome proliferator-activated receptor gamma-coactivator 1-alpha; Drp-1-dynamin-related protein 1; PARP-1—poly(adenosine 5′-diphosphate–ribose) polymerase-1; AP-1- activator protein 1, NF-κB—nuclear factor kappa B; H3—histone H3; HATs/HDACs—histone acetyltransferases/histone deacetylases; GSK-3β—glycogen synthase kinase 3 beta; Tau—microtubule-associated protein; Cer—ceramide; S1P—sphingosine-1-phosphate; SphKs—sphingosine kinases Note that sphingosine kinases were abbreviated as Sphks. However, ASN downregulates the Sphk1 [31], while inhibition of HDACs concerns the S1P synthesized by Sphk2 isoform.

**Figure 3 ijms-22-06277-f003:**
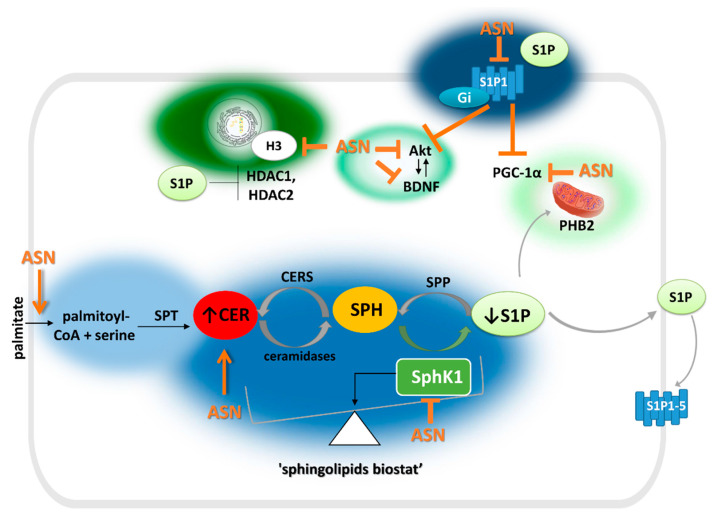
ASN toxicity on S1P metabolism and action. Orange arrows terminating with → (activation), or ┴ (inhibition/deterioration) mark alpha-synuclein (ASN) toxic effects. This includes promotion in palmitate uptake [84], which may increase ceramide synthesis (bright blue), inhibition of sphingosine kinase 1 (SphK1) [31,80,85] (regular blue), as well as expulsion of sphingosine-1-phosphate (S1P) receptor (S1P1) from lipids rafts, and suppression of inhibitory G (Gi) protein [30,80] (navy blue). ASN acts in opposition with S1P in an intracellular and receptor-dependent effect on mitochondrial homeostasis (bright green), including inhibition/up-regulation of peroxisome proliferator-activated receptor gamma-coactivator 1-alpha (PGC-1α) [76,86]. S1P may also regulate assembly of complex IV of respiratory chain by interaction with mitochondrial protein prohibitin 2 (PHB2) [87]. Pathogenic ASN and S1P inhibit and activate, respectively, the Akt kinase (serine/threonine protein kinase B)/brain-derived neurotrophic factor (BDNF) signaling pathway [81,83,88,89] (regular green). S1P inhibits histone deacetylase 1 and 2 (HDAC1 and HDAC2), thus enhancing histone acetylation [90], while ASN hinders access of histone acetyltransferases (HATs) by interacting with histone H3 [28] (dark green). Remaining abbreviations: CoA—coenzyme A; SPT—serine palmitoyl transferase; CER—ceramide; SPH—sphingosine; CERS—ceramide synthase; SPP—S1P phosphatase; S1P1–5—sphingosine-1-phosphate receptors. Please note that sphingosine kinase was abbreviated Sphk1, which is an isoform down-regulated by ASN. However, S1P acting intracellularly on PHB2 and HDACs is synthesized by Sphk2 isoform [90].

**Figure 4 ijms-22-06277-f004:**
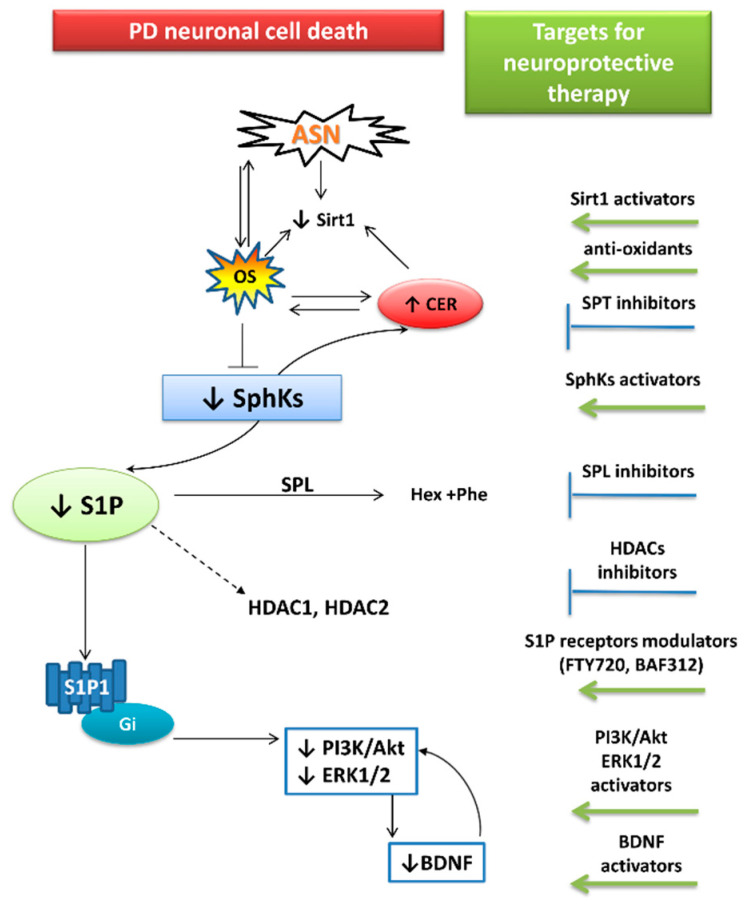
Potential targets for PD neuroprotective strategies associated with sphingolipid signaling and anti-oxidative defense. Arrows terminating with → represent activation, while those terminating with ┴ represent inhibition/deterioration. Abbreviations: PD-Parkinson’s disease; ASN—alpha-synuclein; Sirt1—silence information regulator 1 (sirtuin 1); OS—oxidative stress; CER—ceramide; S1P—sphingosine-1-phosphate; SPL—S1P lyase; Hex + Phe—hexadecenal and ethanolamine phosphate (S1P degradation products); S1P1—S1P receptor 1; Gi—inhibitory G protein; HDAC1, HDAC2—histone deacetylases 1 and 2; PI3K/Akt—phosphatidylinositol 3-kinase/serine/threonine protein kinase B; ERK1/2—extracellular signal-regulated kinases; BDNF—brain-derived neurotrophic factor; FTY720—fingolimod; BAF312—siponimod; SphKs—sphingosine kinases Note that sphingosine kinases were abbreviated as Sphks. However, inhibition under OS conditions mainly affects the Sphk1 isoform synthesizing the S1P pool acting through S1P receptors [9], while Sphk2 synthesizes S1P-inhibiting HDACs. The dashed line connecting S1P (↓S1P) with HDAC1, HDAC2 determines that S1P is an endogenous inhibitor of HDACs, and its decline partially abolishes HDACs inhibition while leading to increased histone deacetylation [90].

**Figure 5 ijms-22-06277-f005:**
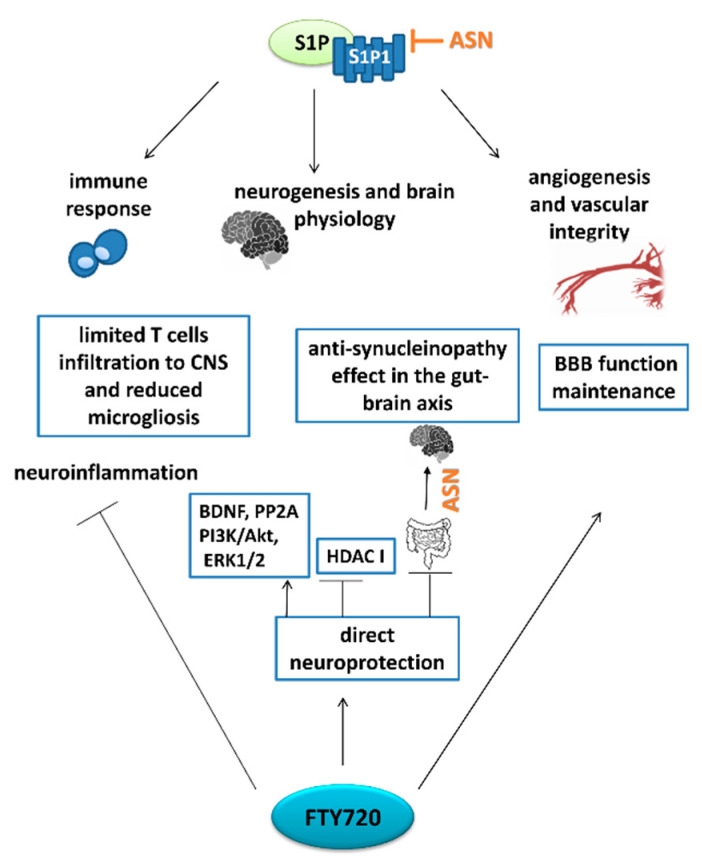
S1P and FTY720 targets of action in neurodegeneration. Arrows terminating with → represent activation, while those terminating with ┴ represent inhibition/deterioration. Sphingosine-1-phosphate (S1P) acts at the stage of embryonic development during angiogenesis and neurogenesis [113], and regulates multiple organ system functions in a mature organism, including cardiovascular, nervous, and immune systems [15,99,125]. ASN may impair S1P1 signaling [80]. Fingolimod (FTY720) targets both the peripheral immune system [170] as well as exerts a direct neuroprotective effect [189], including synucleinopathy reduction in gut-brain axis [64] and also improves the blood–brain barrier (BBB) function [190]. The effects of the FTY720 are shown in blue rectangles. Note that FTY720 direct neuroprotective effect is often independent of known downstream signaling pathways of S1P receptors. Abbreviations: ASN—alpha-synuclein; S1P1–5—sphingosine-1-phosphate receptors; HDAC I—histone deacetylases classes I; BDNF—brain-derived neurotrophic factor; PP2A—protein phosphatase 2A; PI3K/Akt—phosphatidylinositol 3-kinase/serine/threonine protein kinase B; ERK1/2—extracellular signal-regulated kinases.

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
