# Peer review of "Recent Insights into the Interplay of Alpha-Synuclein and Sphingolipid Signaling in Parkinson’s Disease"

_ijms, 2021, doi:10.3390/ijms22126277_

Round 1
Reviewer 1 Report
We appreciate the successful summary of the current knowledge about the interplay of alpha synuclein and sphingolipid signaling in Parkinson´s disease and recommend the publication after following minor revisions.
Minor spelling and grammar:
- Do not use a comma before “and” and “as well as” in general. It sometimes impedes the reading and the comprehensibility of the text
- Recheck the splitting of words at the end of a line
- Line 17: number of evidence - a number of evidence
- Line 26/ 401: receptors modulators - receptor modulators
- Line 46: oxidative stress – OS
- Line 56: to slowing – to slow down
- Line 57: such limiting – such as limiting
- Line 63: be valuable – be a valuable
- Line 70/ 454/ 672 …: SNCA – SNCA; human genes have to be written in italics
- Line 73: Be essential – be an essential
- Line 87: synapses, acting – synapses and acts (currently misleading, not all functions are mediated by the synapses)
- Line 140 – 142: Content difficult to understand; do you want to describe “are characterized by a parallel appearance of motor and non-motor symptoms”.
- Line 147: “also” is repetitive
- Line 224: evokes – evoke
- Line 244: DAT – the abbreviation is not used further in the text
- Line 260: Introduce the abbreviation PHB2 in the main text
- Line 271: opposes – oppose
- Line 405: toxic effects
- Line 449: ANS – ASN
- Line 460: as with PD – as well as
- Line 498: S1P lyase – SPL
- Line 499: grammar difficult, delete “that” and “the comma”
- Line 581: ,also – and also
- Line 698 and Figure 3: Please use consistently either “biostat” or “rheostat”
Content:
- Line 34 – 37: PD follows AD as the most prevalent neurodegenerative disorder. Thus, the first sentence is confusing. If you tried to emphasize, that PD – in contrast to AD - is characterized by motor and non-motor symptoms, please highlight it in the text
- Line 44: Please explain more precisely
- Line 66: Please be more precisely with the chapters. However, we suggest a further splitting of the chapters, which is described in more detail under “Structure”
- Line 101-102: Please add citation
- Line 143: Add citation 43 for the ENS already here
- Line 152: The grammar of this sentence is probably misleading. We don´t know, when the aSyn pathology really starts. However, at the time point of clinical manifestation, already up to 70 % of DA neuron are degenerated. Thus, it is not “at a point when it begins”.
- Line 170: directly name the A53V mutation
Structure:
- For an easier legibility, we would suggest to further split some paragraphs and add additional headlines. E.g. the first paragraph could be split into blocks like “physiological functions of aSyn” and “the role of aSyn in neurodegeneration and PD” and “mechanisms for the toxicity of aSyn in neurodegeneration”
- Line 109 – 127 or 509 or 615: The text contains several content-related repetitions, which can be summarized for a much straighter structure. As an example, this paragraph (line 109 ff) starts with describing inclusions, which is repeated in line 119, after already describing pathological spreading. This impedes a straight structure of the text and could easily be solved. Alike, the first paragraph describing FTY720 should be included into the corresponding chapter and so on.
Figures:
- Figure 2: Explanation of the dashed line is missing
- Line 290: Since a basic knowledge about the sphingolipid rheostat is essential for a good understanding of the complex relations described in this review, we suggest to link this part of text to an additional figure or to Figure 3 and to highlight the basic sphingolipid rheostat mechanism within Figure 3 e.g. with a colored background.
- Figure 3: We (optionally) suggest to add a sub-labeling (1,2,3…) of the specific mechanisms into the figure and to cite/ include them into the respective part of the main text, for an easier association of the main text to the respective graphical representations
- Figure 4: We (optionally) suggest to adapt Figure 4 to a simplified version of Figure 3 including the additional information of Figure 4. This would facilitate the association between previously described/ introduced mechanisms and the presented treatment strategies
- Figure 4: Please make sure to mention all treatment strategies in the paragraph “6. pharmacological targets…”. This does not seem to be the case for “SIRT1 activators” and “anti-oxidants”. If you do not want to re-mention these aspects, please make a note regarding Figure 4 in the corresponding previous section containing the respective information.
- Figure 5: The figure does not comprehensively present FTY720 as S1P receptor modulator.
Author Response
Responses to Reviewer 1 comments
Grammar and content
We corrected the manuscript taking into account all grammatical and stylistic corrections sent by the reviewer ( starting from line 17-till line 698).
Reply to the remarks below:
Reviewer 1
- Line 34 – 37: PD follows AD as the most prevalent neurodegenerative disorder. Thus, the first sentence is confusing. If you tried to emphasize, that PD – in contrast to AD - is characterized by motor and non-motor symptoms, please highlight it in the text
We agree, and for the clarity, we have removed thee phrase ‘the most common’ in the first sentence and the beginning of this chapter now sounds as follows: ‘Parkinson’s disease (PD) is a progressive neurodegenerative disorder characterized by both motor and non-motor disturbances. It is currently accepted as a disease of the whole brain, not only the dopaminergic (DA) system. PD follows Alzheimer’s disease (AD) as the most prevalent neurodegenerative disorder worldwide, with an increasing incidence rate.’
- Line 152: The grammar of this sentence is probably misleading. We don´t know, when the aSyn pathology really starts. However, at the time point of clinical manifestation, already up to 70 % of DA neuron are degenerated. Thus, it is not “at a point when it begins”.
We agree with this remark and therefore we have replaced the previous sentence which was ‘ The classical movement-related PD symptoms appear only a few years or later from the onset of ASN pathology, at a point when it begins to affect the brain regions responsible for motor function with the following sentence ‘The classical movement-related PD symptoms appear only a few years or later from the onset of non-motor signs, and the dynamic of ASN pathology spread from nigrostriatal to widespread cortical and brainstem regions responsible for motor function may be crucial (Barker & Williams-Gray, 2016)
We have also added this new reference.
Moreover, all additional Reviewer 1 suggestions were accepted and included into the manuscript text.
Structure
Reply to the remarks below
Reviewer 1
- For an easier legibility, we would suggest to further split some paragraphs and add additional headlines. E.g. the first paragraph could be split into blocks like “physiological functions of aSyn” and “the role of aSyn in neurodegeneration and PD” and “mechanisms for the toxicity of aSyn in neurodegeneration”
We agree with the reviewer that Chapter 2 needs to be further subdivided, and therefore we have split it into further chapters dealing with the role of ASN in physiology, pathology including Parkinson's disease and finally, ASN signaling protein interaction network.
Furthermore, to remove repetitions and organize the information regarding the ASN-S1P relationship, we moved the section on differences in intracellular ASN and S1P signaling from Chapter 2 to Chapter 4.
- Line 109 – 127 or 509 or 615: The text contains several content-related repetitions, which can be summarized for a much straighter structure. As an example, this paragraph (line 109 ff) starts with describing inclusions, which is repeated in line 119, after already describing pathological spreading. This impedes a straight structure of the text and could easily be solved. Alike, the first paragraph describing FTY720 should be included into the corresponding chapter and so on.
In line with the reviewer's remark, we moved the fragment of the text about synucleinopathy diseases directly after the point where we first write about Lewy's bodies.
figures
Figure 2. (including figure description)
Reply to the remark below
- Figure 2: Explanation of the dashed line is missing
1) The following information has been added regarding the dashed line.
‘A dashed line between ASN and HATs and a little arrow from H3 to HATs / HDACs mean that ASN inhibits histone acetylation by directly associating with histone H3, which masks residues required for acetylation (Kontopoulos et al., 2006). Therefore indirectly, ASN can perturb the HATs / HDACs balance on the side of acetylation inhibition.’
2) We also added the Sphks information in the legend as follows.
‘Note that sphingosine kinases were abbreviated as Sphks. However, ASN downregulates the Sphk1 (Gąssowska et al., 2011; Motyl et al., 2018; Zhang et al., 2017), while inhibition of HDACs concerns the S1P synthesized by Sphk2 isoform (Hait et al., 2009).’
Similar information is also included in Figure 3 and Figure 4.
3) The following information has also been added regarding the colours of the proteins that interact with the ASN.
‘Proteins interacting with ASNs have been grouped and colour coded into categories depending on their biological function.’
4) We simplified figure 3 by removing nitric oxide synate (NOS) in the sphingolipd biostat category. Instead, we added the S1P / ceramide imbalance box. We also split HDACs and HATs into 2 boxes to clearly show the different effect of S1P and ASN on both enzymes
Figure 3. (including figure description)
- Line 290: Since a basic knowledge about the sphingolipid rheostat is essential for a good understanding of the complex relations described in this review, we suggest to link this part of text to an additional figure or to Figure 3 and to highlight the basic sphingolipid rheostat mechanism within Figure 3 e.g. with a colored background.
- Figure 3: We (optionally) suggest to add a sub-labeling (1,2,3…) of the specific mechanisms into the figure and to cite/ include them into the respective part of the main text, for an easier association of the main text to the respective graphical representations
1) According to the reviewer suggestion, We marked with different shades of colour the effect of ASN on metabolism and S1P action described in the legend.
2) We simplified Figure 3 so that it shows the role of the toxic ASN, according to its title.
3) We changed the sphingosine kinase abbreviation from Sphks to Sphk1 as this isoform is down-regulated by ASN. We also added the appropriate information in the legend.
Figure 4. (including figure description)
Reviewer 1
- Figure 4: Please make sure to mention all treatment strategies in the paragraph “6. pharmacological targets…”. This does not seem to be the case for “SIRT1 activators” and “anti-oxidants”. If you do not want to re-mention these aspects, please make a note regarding Figure 4 in the corresponding previous section containing the respective information.
We've added a brief overview of the importance of Sirtuin 1 and its anti-oxidant protection in Chapter 6.
Figure 5.
Reply to the remarks below
Reviewer 1
- Figure 5: The figure does not comprehensively present FTY720 as S1P receptor modulator.
We wish to point out that it was not our intention to demonstrate fingolimod (FTY720) as a modulator of S1P receptors. In addition to the well-known mechanism of FTY720 action dependent on the S1P1 receptor on lymphocytes, the drug may also have a direct effect, for example, on HDACs (Hait et al., 2009). Moreover, in Alzheimer's disease model, FTY720-mediated amyloid-beta reduction was independent of known downstream signaling pathways of S1PRs (Takasugi et al., 2013).
The following information has been added on legend
‘Note that direct neuroprotective effect is often independent of known downstream signaling pathways of S1PRs.’
Additionally In chapter 6 we write
‘The mechanism of FTY720 action is both S1P-receptor-dependent, as well as mimicking the intracellular action of S1P. One of the crucial findings in this field was that FTY720 enters nucleus and, after SphK2-mediate phosphorylation, inhibits HDAC class I (Hait et al., 2014). It was shown to inhibit hippocampal HDAC enzymatic activity in severely immunodeficient mice and to enhance
acetylation of histone lysine residues connected with the epigenetic regulation of learning and memory genes’
Literature:
Barker, R. A., & Williams-Gray, C. H. (2016). The spectrum of clinical features seen with alpha synuclein pathology. Neuropathology and Applied Neurobiology, 42(1), 6–19. https://doi.org/10.1111/nan.12303
Gąssowska, M., Każmierczak, A., Strosznajder, R. P., Adamczyk, A., & Strosznajder, J. B. (2011). Sphingosine kinase(s) and their role in alpha-synuclein secretion and toxicity. Neurochemical Conference, Warsaw, Poland, 63(5), 1288–1289. https://doi.org/10.1016/S1734-1140(11)70677-6
Hait, N. C., Allegood, J., Maceyka, M., Strub, G. M., Harikumar, K. B., Singh, S. K., Luo, C., Marmorstein, R., Kordula, T., Milstien, S., & Spiegel, S. (2009). Regulation of histone acetylation in the nucleus by sphingosine-1-phosphate. Science (New York, N.Y.), 325(5945), 1254–1257. https://doi.org/10.1126/science.1176709
Hait, N. C., Wise, L. E., Allegood, J. C., O’Brien, M., Avni, D., Reeves, T. M., Knapp, P. E., Lu, J., Luo, C., Miles, M. F., Milstien, S., Lichtman, A. H., & Spiegel, S. (2014). Active, phosphorylated fingolimod inhibits histone deacetylases and facilitates fear extinction memory. Nature Neuroscience, 17(7), 971–980. https://doi.org/10.1038/nn.3728
Kontopoulos, E., Parvin, J. D., & Feany, M. B. (2006). α-synuclein acts in the nucleus to inhibit histone acetylation and promote neurotoxicity. Human Molecular Genetics, 15(20), 3012–3023. https://doi.org/10.1093/hmg/ddl243
Motyl, J., Wencel, P. L., Cieślik, M., Strosznajder, R. P., & Strosznajder, J. B. (2018). Alpha-synuclein alters differently gene expression of Sirts, PARPs and other stress response proteins: implications for neurodegenerative disorders. Molecular Neurobiology, 55(1), 727–740. https://doi.org/10.1007/s12035-016-0317-1
Takasugi, N., Sasaki, T., Ebinuma, I., Osawa, S., Isshiki, H., Takeo, K., Tomita, T., & Iwatsubo, T. (2013). FTY720/Fingolimod, a Sphingosine Analogue, Reduces Amyloid-β Production in Neurons. PLoS ONE, 8(5), 1–8. https://doi.org/10.1371/journal.pone.0064050
Van Brocklyn, J. R., & Williams, J. B. (2012). The control of the balance between ceramide and sphingosine-1-phosphate by sphingosine kinase: Oxidative stress and the seesaw of cell survival and death. Comparative Biochemistry and Physiology - B Biochemistry and Molecular Biology, 163(1), 26–36. https://doi.org/10.1016/j.cbpb.2012.05.006
Zhang, L., Okada, T., Badawy, S. M. M., Hirai, C., Kajimoto, T., & Nakamura, S. (2017). Extracellular α-synuclein induces sphingosine 1-phosphate receptor subtype 1 uncoupled from inhibitory G-protein leaving β-arrestin signal intact. Scientific Reports, 7(1), 44248. https://doi.org/10.1038/srep44248

Reviewer 2 Report
In this review, the authors summarize the toxicity of ASN and Sph signaling very well. Minor points that should be corrected are given below.
Minor points,
1. It is difficult to understand whether the interactions and functions of ASN shown in the main text and figures are about normal ASN, toxic ASN, or both. In particular, I think it would be easier to distinguish between normal ASN and toxic ASN by using different colors in the figures.
2. The meaning of the dashed line in Figure 2 is not indicated.
3. Figure 4 is difficult to understand. It is hard to tell whether each is inhibiting or promoting, because what is supposed to be inhibiting is marked as decreasing. (e.g. OS inhibit SphKs decreasing?)
Author Response
Responses to Reviewer 2 comments
We corrected the manuscript taking into account all Reviewer suggestions.
Reply to the remarks below:
- The meaning of the dashed line in Figure 2 is not indicated.
1) The following information has been added regarding the dashed line.
‘A dashed line between ASN and HATs and a little arrow from H3 to HATs / HDACs mean that ASN inhibits histone acetylation by directly associating with histone H3, which masks residues required for acetylation (Kontopoulos et al., 2006). Therefore indirectly, ASN can perturb the HATs / HDACs balance on the side of acetylation inhibition.’
2) We also added the Sphks information in the legend as follows.
‘Note that sphingosine kinases were abbreviated as Sphks. However, ASN downregulates the Sphk1 (Gąssowska et al., 2011; Motyl et al., 2018; Zhang et al., 2017), while inhibition of HDACs concerns the S1P synthesized by Sphk2 isoform (Hait et al., 2009).’
- Figure 4 is difficult to understand. It is hard to tell whether each is inhibiting or promoting, because what is supposed to be inhibiting is marked as decreasing. (e.g. OS inhibit SphKs decreasing?)
1) We replaced the inhibition arrow with the dashed line connecting S1P (low level) with HDAC1, HDAC2. In this way, we show that S1P is an endogenous inhibitor of HDACs and its reduction partially abolishes HDACs inhibition while leading to increased histone deacetylation.
2) The following information has also been added regarding Sphks inhibition under oxidative stress (OS) conditions.
Note that sphingosine kinases were abbreviated as Sphks. However, inhibition under OS conditions mainly affects the Sphk1 isoform (Van Brocklyn & Williams, 2012). We also added information on the potential pro-apoptotic properties of Sphk2 in Chapter 3. S1P Metabolism and its Role in PD
3) We have unified the Sirt1 notation (instead of SIRT1).
Reply to Reviewer 2 general comments
- It is difficult to understand whether the interactions and functions of ASN shown in the main text and figures are about normal ASN, toxic ASN, or both. In particular, I think it would be easier to distinguish between normal ASN and toxic ASN by using different colors in the figures.
We agree with the reviewer that Chapter 2 needs to be further subdivided, and therefore we have split it into further chapters dealing with the role of ASN in physiology, pathology including Parkinson's disease and finally, ASN signaling protein interaction network.
Furthermore, to remove repetitions and organize the information regarding the ASN-S1P relationship, we moved the section on differences in intracellular ASN and S1P signaling from Chapter 2 to Chapter 4.
We have marked the ASN in the figures from 2 till 5 in orange (pathological function). However, we would like to point out that this is a considerable simplification because it is impossible to clearly separate when ASN assumes a physiological function and when a pathological one, and both these functions often overlap and in the cell the end result is observed which results from the advantage of one form over the other. Instead of using the term toxic ASN (in the main text), we usually use the term ASN overexpression or mutation or extracellular ASN, because in experimental models this results in a pathological accumulation of ASN that can lead to toxicity of this protein similar to Parkinson's Disease.
Literature:
Gąssowska, M., Każmierczak, A., Strosznajder, R. P., Adamczyk, A., & Strosznajder, J. B. (2011). Sphingosine kinase(s) and their role in alpha-synuclein secretion and toxicity. Neurochemical Conference, Warsaw, Poland, 63(5), 1288–1289. https://doi.org/10.1016/S1734-1140(11)70677-6
Hait, N. C., Allegood, J., Maceyka, M., Strub, G. M., Harikumar, K. B., Singh, S. K., Luo, C., Marmorstein, R., Kordula, T., Milstien, S., & Spiegel, S. (2009). Regulation of histone acetylation in the nucleus by sphingosine-1-phosphate. Science (New York, N.Y.), 325(5945), 1254–1257. https://doi.org/10.1126/science.1176709
Kontopoulos, E., Parvin, J. D., & Feany, M. B. (2006). α-synuclein acts in the nucleus to inhibit histone acetylation and promote neurotoxicity. Human Molecular Genetics, 15(20), 3012–3023. https://doi.org/10.1093/hmg/ddl243
Motyl, J., Wencel, P. L., Cieślik, M., Strosznajder, R. P., & Strosznajder, J. B. (2018). Alpha-synuclein alters differently gene expression of Sirts, PARPs and other stress response proteins: implications for neurodegenerative disorders. Molecular Neurobiology, 55(1), 727–740. https://doi.org/10.1007/s12035-016-0317-1
Van Brocklyn, J. R., & Williams, J. B. (2012). The control of the balance between ceramide and sphingosine-1-phosphate by sphingosine kinase: Oxidative stress and the seesaw of cell survival and death. Comparative Biochemistry and Physiology - B Biochemistry and Molecular Biology, 163(1), 26–36. https://doi.org/10.1016/j.cbpb.2012.05.006
Zhang, L., Okada, T., Badawy, S. M. M., Hirai, C., Kajimoto, T., & Nakamura, S. (2017). Extracellular α-synuclein induces sphingosine 1-phosphate receptor subtype 1 uncoupled from inhibitory G-protein leaving β-arrestin signal intact. Scientific Reports, 7(1), 44248. https://doi.org/10.1038/srep44248
